# Has a High Dose of Vitamin D3 Impacted Health Conditions in Older Adults?—A Systematic Review and Meta-Analysis Focusing on Dose 100,000 IU

**DOI:** 10.3390/nu16020252

**Published:** 2024-01-14

**Authors:** Barbara Owczarek, Anna Ziomkiewicz, Edyta Łukowska-Chojnacka

**Affiliations:** 1Chair of Drug and Cosmetics Biotechnology, Faculty of Chemistry, Warsaw University of Technology, ul. Noakowskiego 3, 00-664 Warsaw, Poland; edyta.chojnacka@pw.edu.pl; 2Research and Development Department, Pharmaceutical Works Polpharma S.A. Medana Branch in Sieradz, ul. Łokietka 10, 98-200 Sieradz, Poland; 3Industrial Operations Quality Assurance Department, Pharmaceutical Works Polpharma S.A. Medana Branch in Sieradz, ul. Łokietka 8, 98-200 Sieradz, Poland; 4Laboratory of Anthropology, Institute of Zoology and Biomedical Research, Faculty of Biology, Jagiellonian University, ul. Gronostajowa 9, 30-387 Kraków, Poland; anna.ziomkiewicz-wichary@uj.edu.pl

**Keywords:** vitamin D3, vitamin D3 deficiency, high doses, older people, systematic review

## Abstract

Background: Older adults are prone to vitamin D3 (VD3) deficiency, which may impair their health. A high dose of VD3 (HDVD3 = 100,000 IU) could improve their 25-hydroxyvitamin D3 [25(OH)D] level and health outcomes. However, evidence for such a beneficial effect of HDVD3 in older adults coming from clinical trials is mixed. Objective: To review the literature on the efficacy of a single dose of 100,000 IU of VD3 in older people. Methods: We searched PubMed/Medline, Science Direct, and NIH’s clinical trials registry for clinical studies on the effect of a single high dose of VD3 on various health outcomes in older people. We also performed a meta-analysis using the standardized mean difference to assess the effect of VD3 on its blood level. Due to expected high heterogeneity, its amount (i.e., tau^2^) was estimated using the DerSimonian-Laird estimator. To estimate tau^2^, the Q-test for heterogeneity and the *I*^2^ statistic were calculated. Results: Search results identify 13 studies that reported diverse health outcomes, such as lung and cardiovascular function, skin cancer progression, intensive care unit mortality, immune system response, and bone density. The meta-analysis showed a significant increase in 25(OH)D blood levels after treatment in 10 studies, with an average standardized mean difference of 2.60 ng/mL (95% CI: 2.07 to 3.13). Their results suggested that a single high dose of VD3 may benefit intensive care unit patients and skin cancer patients in remission. However, evidence for other beneficial health effects of HDVD3 was mixed due to high heterogeneity among studies. Conclusions: A single high dose of VD3 may positively affect some health outcomes in older people, possibly due to its pleiotropic and immunomodulatory effects. However, the evidence needs to be more extensive and consistent, and more rigorous studies are required to confirm the benefits and safety of VD3 high doses in older patients.

## 1. Introduction

Vitamin D3 (cholecalciferol, VD3) is a fat-soluble vitamin that contributes to maintaining calcium-phosphate homeostasis and plays a crucial role in several other biological processes, such as neuromuscular function, immune system function and reduction of inflammation, glucose metabolism or modulation processes of cell growth [1,2,3]. VD3, in its active form, 25-hydroxyvitamin D3 [25(OH)D], also modulates many genes in our body, which encode proteins responsible for regulating cell proliferation, differentiation, and apoptosis [1]. Due to insufficient dietary intake and to maintain its proper level in the organism, supplementation is recommended. Since 2010, taking up dedicated doses of VD3 per day, at most 2000 IU, has been suggested [2]. A higher dose of VD3, e.g., 6000 IU/day, is acceptable for children over one year old in rickets treatment. However, higher doses should always be adjusted to the patients’ 25(OH)D blood serum level [3].

Currently, many pharmaceutical companies are marketing products containing high doses of cholecalciferol. Available preparations on the market contain 10,000 IU, 20,000 IU, 50,000 IU, or even 100,000 IU of vitamin D3. However, the safety and efficacy of the treatment with high doses remain unresolved, especially in older adults. Although high doses are available to patients with a doctor’s prescription, it is still being determined if their administration is always clinically justified.

The use of very high doses of VD3 still raises many questions about their validity and safety. Few studies are strictly related to doses above 100,000 IU. Using such high doses may increase the blood concentration of 25(OH)D to the optimal level, which is an argument in favor of such doses. Moreover, using such doses prophylactically at longer intervals, e.g., once a month or even once every three months, is more convenient for the patient. The dose is available in many forms, e.g., for injection, in ampoules for oral administration, or soft gel capsules.

As for the safety and quality of available preparations, there is a risk when persons self-administer high doses, which may cause side effects. There are also clinically untested preparations advertised on websites containing much higher doses of VD3—e.g., 200,000 IU, which the patient can buy online. However, these specifics do not have the status of a medicinal product but a dietary supplement [4,5]. Therefore, it is reasonable to attempt to systematise the validity of using a dose of 100,000 IU.

Beneficial effects of high doses of vitamin D3 (HDVD3) were noticed in respiratory tract-associated diseases [6,7], but for bone density [8,9], the outcomes are only sometimes clear, especially in older adults. Identifying disease entities for which HDVD3 will bring measurable benefits in older people is also problematic. Thus, during this systematic review and meta-analysis, we focus on answering whether there is any clear impact of administering high doses of VD3 on improving general health conditions in the elderly compared with no treatment or treatment with lower doses. From 2015 to 2050, the elderly population (over 60 years old) is expected to increase from 12% to 22%, considering the world population [10,11], making this issue increasingly important.

Scientific research shows that a proper level of 25(OH)D in serum maintains appropriate mineral balance and reduces the risk of diseases such as type 2 diabetes [11,12,13] or depression [14,15,16]. Moreover, VD3 treats many diseases, such as osteomalacia in adults [17], thyroid and autoimmunological diseases [18,19], rheumatoid arthritis, and Hashimoto disease. Also, since the onset of the COVID-19 pandemic, clinical trials have been conducted on the effects of VD3 on the course of the disease among patients who take high doses of VD3 [8,20,21,22,23]. Additionally, the lower dose of VD3 (600–4000 IU) is used in prevention as a dietary supplement.

Achieving the level of 25(OH)D in the serum, which impacts health, depends on many factors [24], including age, body weight, obesity [25], pigmentation, comorbidities, and genetics. These factors determine the rate of VD3 metabolism [26,27], tissue distribution, and pharmacokinetics. Thus, the effect of high doses of VD3 has also been investigated.

Age is a significant risk factor contributing to 25(OH)D deficiency in older adults due to reduced intake of VD3 in food [28]. Senior people often experience difficulties in eating, which may result from problems with decreased appetite, swallowing, dentition, medication usage, or a history of illness (stroke, Parkinson’s disease). Additional factors associated with age are being overweight [25], polypharmacy (taking many medicines), decreased cutaneous synthesis of 25(OH)D because of a short time of outdoor activities, and full-body covering with clothes [29,30]. Older adults are often hospitalized and require care, including special nutrition or dedicated medicines. Ensuring the appropriate intake of all necessary dietary ingredients is challenging in such conditions and may significantly impact health in older adults.

An appropriate serum level of 25(OH)D for people over 75 years old can be helpful in the treatment of some diseases mentioned above [15,17,18,24,28,29]. In the investigations conducted in 2016 [30,31,32], high doses (50,000 IU or 1,000,000 IU) of VD3 were administered once a week to patients with statin intolerance and with low serum levels of 25(OH)D. The researchers established the safety of 50,000 to 100,000 IU per week of vitamin D3 supplementation for up to 1 year [30,31,32]. A similar conclusion was made in a study by Hossein-Nezhad and Holick, where they found that daily doses of vitamin D3 up to 10,000 IU were safe in healthy males [33]. Reports from another clinical trial [7] suggest that using even higher doses of cholecalciferol in the order of several dozen higher given in recommendations or even 100,000 IU within a single administration is safe for randomized patients. No evidence of VD3 toxicity has been reported in healthy adults who received 50,000 IU of VD3 once every two weeks (equivalent to approximately 3300 IU/day) in a clinical setting for up to 6 years [34]. In a long-term study conducted by McCullough et al. [35], high doses of VD3 (5000–50,000 IU/day or even 50,000–100,000 IU/day) were also administered to hospitalized patients. This seven-year study demonstrated that such high doses appear to be safe and do not cause an increase in parathormone or hypercalcemia. Moreover, there was no observed toxicity of even 20,000 IU of VD3 delivered daily in Canadian adult patients if individuals administered it according to the physician’s recommendation. The 25(OH)D level in the serum of these patients increased significantly, even up to 60 ng/mL (150 nm/L) [34].

Considering the heterogeneity in the available clinical evidence regarding the effect of HDVD3 on the health of older patients, the present systematic review and meta-analysis were undertaken to provide a more accurate summary and collate the effect of VD3, mainly focusing on a dose of 100,000 IU on clinical outcomes in older adults. Researchers also considered the studies’ heterogeneity and potential limitations during this systematic review. The age of the patients included in the reviewed studies ranged from 50 to 85, which suggests that the heterogeneity of this group was significant. Concomitant diseases, different initial levels of 25(OH)D in the serum, general health condition, BMI (or body weight), gender, and number of drugs and supplements used mean that drawing clear conclusions should be cautiously approached.

## 2. Materials and Methods

This meta-analysis was conducted and reported according to the Preferred Reporting Items for Systematic Reviews and Meta-analyses (PRISMA) statement [36]. The investigated question was created using the PICO methodology. The defined population included older adults (age 50–96), women, and men with accompanying diseases such as hypertension, diabetes, etc. The intervention was orally or intravenously administered 100,000 IU of VD3. The comparison was placebo, no treatment, or therapy with smaller doses. The outcome was a positive or negative result, whereas the negative result included no changes after administration of VD3 in the expected area of life or health.

In summary of the PICO model, the main question was stated as follows: In older adults (50+) with other diseases (both women and men), is the administration of 100,000 IU of VD3 (orally or intravenously) more effective in producing clear outcomes, compared to placebo, no treatment, or lower- dose therapy, or can it lead to negative consequences, including no changes in the investigated population’s health status?

The study protocol was registered in PROSPERO (Registration number CRD42022337348) [37].

### 2.1. Selection of Studies

Studies were regarded as eligible if they were: (1) observational studies (retrospective, cohort, or case-control design) as well as randomized controlled trials (RCT) irrespective of study design (parallel/cross-over), study blinding (single-blind, double-blind, or open-label), and sample size would be included in the meta-analysis; (2) studies included adult patients, especially elderly patients, a proportion of whom were taking VD3 in high doses (100,000 IU) irrespective of the dose, duration, or formulation of VD3 used; (3) studies reported clinical outcomes of elderly patients with low 25(OH)D serum; (4) the clinical outcomes were reported in patients with VD3 supplementation compared to those who did not receive VD3 or patients taking a placebo; and (5) results of studies were published 2015 or later.

Exclusion criteria were as follows: (1) studies reporting clinical outcomes of pediatric patients and clinical outcomes of adult patients for cholecalciferol dose lower than 100,000 IU; (2) comments, editorials, letters to the editor, encyclopedia, book chapters, conference reports, discussions, errata, news, data articles, correspondence; (3) non-peer reviewed studies published as preprints; or (4) incompleteness of data.

### 2.2. Data Sources and Search Strategy

We independently searched databases of the literature across PubMed/Medline, NIH’s clinical trials registry [www.clinicaltrials.gov (accessed on 15 May 2022)], and Science Direct to identify relevant studies. Using the following keywords, interposed with appropriate Boolean operators: “vitamin D high doses elderly”, “cholecalciferol high doses elderly”, “cholecalciferol 100,000 IU elderly”, “vitamin D 100,000 IU elderly”, “cholecalciferol 100,000 IU adults”, “vitamin D3 100,000 IU adults”, “high doses of cholecalciferol”.

The language was restricted to English only. The references of relevant reviews and articles were also screened for potentially eligible articles. For missing data, the corresponding authors of the potentially eligible studies were contacted wherever possible.

### 2.3. Data Extraction and Risk of Bias Assessment

Two investigators (BO and AZ) independently reviewed titles and abstracts to exclude duplicate studies and studies that failed to meet the previously mentioned eligibility criteria. Potentially eligible studies were full-text assessed. Any discrepancies between the investigators were solved by discussion or consensus. Selected studies were reviewed, and the following data were extracted for further assessment: study characteristics, a dose of VD3, time duration of HDVD3 supplementation, formulation and mode of VD3 administration, the number of patients supplemented with VD3, the number of patients with VD3 supplementation who had experienced the reported clinical outcome as compared to those who did not receive VD3 (i.e., the number of events in those supplemented with VD3 vs. those not supplemented).

The reviewers Independently assessed the risk of bias in trials using the Cochrane risk of bias assessment tool [38]. The issue was classified as “high”, “low”, or “unclear” to the following items: random sequence generation, allocation concealment, blinding of participants and personnel, and blinding of outcome assessment, selective reporting, and other sources of bias.

### 2.4. Statistical Method

To assess the overall effect of HDVD3 treatment on the level of VD3 in older study participants, we applied a meta-analytical approach using the standardized mean difference as the outcome measure. A random-effects model was fitted to the data. Due to expected high heterogeneity, its amount (i.e., tau^2^) was estimated using the DerSimonian-Laird estimator) [39]. In addition to the estimate of tau^2^, the Q-test for heterogeneity [40] and the *I*^2^ statistic were calculated. If any amount of heterogeneity was detected (i.e., tau^2^ > 0, regardless of the results of the Q-test), a prediction interval for the true outcomes was also provided. Studentized residuals and Cook’s distances were used to check for outliers and/or influentials in the context of the model. Studies with a studentized residual larger than the 100 × (1 − 0.05/(2 × k))th percentile of a standard normal distribution were considered potential outliers. Studies with a Cook’s distance larger than the median plus six times the interquartile range of the Cook’s distances ’ere considered influential. The rank correlation and regression tests used the standard error of the observed outcomes as a predictor to check for funnel plot asymmetry. Statistical analysis was performed using the JAMOVI software, version 2.3 [41].

## 3. Results

Out of the 1204 available articles, 14 remained potentially eligible after the title and abstract screening, and 13 articles involving 7394 participants were included in the review (Figure 1).

As the studies included in the systematic review concerned various health outcomes, they were grouped into five subcategories depicting the effect of HDVD3 treatment on (1) lung function, (2) cardiovascular system functions (including hypertension), (3) all-cause mortality, (4) immune function, and (5) outcomes connected with bones and falls. One study included the outcomes for the three subgroups mentioned above (cardiovascular disease, acute respiratory infection, falls and non-vertebral fractures) [42].

The flowchart of the study selection process is shown in Figure 1.

### 3.1. Characteristics of the Studies

Of the 13 studies, 12 were randomized clinical studies [7,8,9,42,43,44,45,46,47,48,49], eight included a placebo group [6,25,36,37,38,41,42,43], 7 were double-blinded [6,25,36,37,42,43,50], and one study was quasi-experimental [51]. Selected studies were conducted in the USA [6,42,46], France [41,44], the UK [7], New Zealand [9,43,47,50], Brazil [49], and Italy [45]. A summary of these trials is presented in Table 1 and Table 2. The largest studies were from New Zealand (5108 subjects) and the UK (1901 subjects). In the USA, 42 patients were included in selected studies; in France, 192; in Italy, 104; and in Brazil, 43. The shortest VD3 supplementation period was five days [41,44], while it was 3.3 years [50].

In some studies, patients enrolled in RCTs were on medications during the trial [42,46]. These included corticosteroids, methotrexate, anti-TNF, tocilizumab, abatacept, and rituximab. In the case of Camargo et al. [46], all patients were treated with corticosteroids. In the study by Soubrier et al. [42] 21 patients were on corticosteroids, 43 patients on methotrexate, 49 on biotherapy, including 18 with anti-TNF, 15 patients with tocilizumab, 14 with abatacept and two with rituximab. In COVID-19-associated RCT, corticosteroids and antibiotics were administered [8].

Table 2 presents the period of HDVD administration from the shortest (days) to the longest (years). In each case, there is an increase in the serum concentration of 25(OH)D in the study group compared to the control group, where the level of 25(OH)D did not change significantly. In the group of patients who received HDVD, its level increased on average approximately twice after the examined period. In each case, the study group achieved concentrations within the normal range. 

### 3.2. Risk of Bias

The risk of bias assessments of the included studies is summarized in Table 3. Selected studies were categorized as having high, low, or unclear risks of bias, respectively. Attrition and reporting bias were primary sources of bias.

During risk of bias analysis, we considered random sequence generation, allocation concealment, performance bias including blinding personnel and participants, incompleteness of outcome data, selective reporting, and other biases such as studies authors’ connections with financing or gender allocation for each trial. Random sequence generation was not described [44,49,52]. The highest risk was determined for RCT results described by Soubrier et al. [44], 84% of the sample were women, so the results are difficult to extrapolate to both genders and the general population. These authors did not report the baseline 25(OH)D level or the post-supplementation 25(OH)D level results.

In the RCT by Annweiler et al., no clear indication of allocation concealment and blinding personnel and participants was provided. No 25(OH)D levels post-supplementation were reported for the RCT carried out by Annweiler et al. [52] For the RCT carried out by Camargo et al. [47], no SD for 25(OH)D post-supplementation was reported. The corresponding author confirmed the lack of SD in the RCT [47].

Means and SDs were calculated from median and confidence intervals for three studies [43,46,48]. For another RCT, the population of participants decreased from 52/52 (at the beginning of the study, test/control groups) to 25/22 (after three years) because of excluding some individuals during the test period [9]. Due to changes in the test and control groups’ size, the calculation was based on results after three years (25/22 people in the test/control groups). Characteristics of the included studies are presented in Table 4.

### 3.3. Lung function

The association between VD3 treatment and lung condition was investigated in three RCTs [41,45,47]. In each of these studies, HDVD3 was administered once a month, which may have been an insufficient dose to observe improved lung function in all participants despite the overall increase in serum VD3 levels. Two trials [41,45] focused on lung function (n = 1260 in all studies) reported a change in serum vitamin D3 levels after the intervention. In one RCT (VD3 was given once per month for 3.3 years), no significant lung function improvement (VD3 vs. placebo) was found in the total sample of participants, patients with VD3 deficiency, neither with asthma nor subjects with chronic obstructive pulmonary disease [41]. In another study, VD3 supplementation was administered once per month for 1.1 years, and it improved lung function in ever-smokers, especially those with VD3 deficiency, asthma, or COPD [45]. In patients without smoking episodes, HDVD did not improve lung conditions. A mean level of 25(OH)D increased from 61 (SD = 24) nmol/L at baseline to 119 (SD = 45) nmol/L at follow-up in the supplemented group, whereas in the control group, the serum level did not change [45]. No significant lung function improvement was found in the total sample (VD3 vs. placebo), VD3-deficient, or asthma/COPD participants [45]. The third study found evidence of possible benefit among those patients with severe VD3 deficiency (baseline 25(OH)D < 25 nmol/L). In this trial, VD3 was administered once per month for years [47]. The researchers noticed potential benefits among those patients with vitamin D deficiency. However, RCT indicates no overall impact of HDVD on exacerbations of asthma or COPD in examined elderly adults [47].

### 3.4. Cardiovascular Function

Studies on mortality due to specific diseases indicate that the most common cause of death is heart disease [54]. Cardiovascular diseases are common in older adults; however, the effectiveness of treatment with high doses of vitamin D3 differs depending on the study outcome.

De Paula et al. [38] published the results of the RCT with the primary outcome of improved blood pressure for inpatients with type 2 diabetes, hypertension, or vitamin D3 insufficiency, regardless of normalization of VD3. In this study, a single HDVD3 of 100,000 IU was considered a valuable tool to improve cardiovascular function in patients with type 2 diabetes, hypertension, or hypovitaminosis D [25(OH)D < 20 ng/mL]. Applying a single dose of VD3 resulted in clinically significant decreases in blood pressure. The most relevant effects were observed in ambulatory blood pressure monitoring measurements, and decreases were observed in 24-h systolic (−7.5 mm Hg), daytime systolic (−7 mm Hg), and nighttime systolic (−7 mm Hg) blood pressure. Another trial demonstrated no significant changes in blood pressure during the observation period of 3.3 years (range between 2.5 to 4.2 years) [50]. VD3 in a soft gel capsule was administered monthly during the study period.

### 3.5. Immune Function

Two studies examined the effect of high doses of vitamin D3 on the immune system. Each concerned a different issue, but the results obtained were similar and indicated no improvement in the parameters studied in the study group vs. the placebo group. The first study was related to rheumatoid arthritis (RA), an immune-mediated disease. VD3 has an immunomodulating potential that is helpful as adjuvant therapy in RA treatment. A trial by Soubrier et al. investigated the effect of 100,000 IU of VD3 on improvement in functional disability of patients with RA and VD3 deficiency [42]. In this trial, VD3 was administered once per month for 24 months. The primary outcome was investigated using the Health Assessment Questionnaire (HAQ) [55]. The results were presented with and without adjustment. At six months, without adjustment, there was no significant difference between the VD3 and placebo groups regarding the HAQ score. The HAQ score tended to increase in the placebo group (+0.08 ± 0.25) while slightly diminish in the VD3 group (−0.03 ± 0.23). When factors such as age, gender, season, and initial VD3 status were adjusted, both groups’ differences achieved statistical significance (the VD3 group −0.03 ± 0.23 vs. 0.08 ± 0.25 the placebo group). In this DB RCT, researchers did not observe clinically relevant effects, even in patients with RA and VD3 deficiency. Overall, the quality of life was not improved in the treatment group compared to the placebo group. This may be because the 100,000 IU dose was administered once a month, which equates to 3333 IU per day and may not have been sufficient to see the desired effect.

We also analyzed the results of the second RCT, where a correlation between VD3 and the immune system was investigated [6]. VD3 was given once per 15 days for three months. The researchers observed the immune system response (level of antibodies) and other parameters such as serum cathelicidin, plasma cytokines, lymphocyte phenotyping, and phagocyte ROS production after administering HDVD and the influenza vaccine. The results suggest that VD3 supplementation in deficient elderly persons is ineffective in improving their antibody response to the influenza vaccine, even though 25(OH)D increased significantly.

### 3.6. Bone Density, Falls and Non-Vertebral Fractures

VD3 supplementation, due to its regulatory effect on calcium metabolism, is frequently recommended as a preventive measure in osteoporosis. Several clinical trials have demonstrated the protective effect of lower doses of VD3 on bone mineral density and risk of fractures [56]. The impact of HDVD3 supplementation on bone density in older adults was evaluated in two trials with mixed outcomes [47,52]. In both studies, HDVD3 was used at long intervals—once a month, which recalculates into around 3330 IU daily. It is possible that increasing the frequency of HDVD3 (100,000 IU) administration would result in the preferable outcome.

In the first trial, the authors investigated the association between HDVD3 (100,000 IU VD3 once per month for 3.3 years), bone mineral density (BMD), and non-vertebral fractures (NVF). During this study, serum 25(OH)D concentrations did not change in the placebo group (+1.32 ng/mL) but rose in the VD3 group (+29.2 ng/mL). At two years of treatment, mean 25(OH)D levels were 24 ng/mL (SD 9.5) and 51.6 ng/mL (SD 11.2) in the placebo and VD3 groups, respectively. However, no significant differences in BMD and NVF were found between those groups. The first study indicates a lack of benefit from HDVD3 supplementation on bone and other endpoints when baseline 25(OH)D is above 12 ng/mL. Except for changes in VD3 serum, the researchers did not observe any significant effect of taking 100,000 IU VD3 on bone density. 

The second RCT focused on the impact of HDVD3 on falls and NVF [52]. In this trial, patients were divided into two groups- the intervention group taking 100,000 IU VD3 every 2–3 months for two years and the placebo group. During the study, 2638 participants reported falling, 52% out of 2539 in the VD3 group compared with 53% out of 2517 in the placebo group. NVF was reported in 292 out of 2558 individuals, of whom 6% (156) were in the VD3 treatment group and 5% (136) were in the placebo group. The adjusted non-significant HR for fractures was 1.19 (95% CI 0.94–1.50; *p* = 0.15) for the VD3 compared with the placebo group. The authors concluded that monthly HDVD3 supplementation had no beneficial effect on falls or fractures in the healthy, ambulatory, elderly population.

### 3.7. Cancer-Free Survival

The burden of cancer is significantly increasing compared to previous years among the elderly [36,57,58,59,60], which constitutes a continuous challenge for healthcare systems. Studying the effectiveness of vitamin D3 on improving health in the case of cancer could be a promising strategy. Within the confines of our comprehensive review, we identified a singular study that administered a dosage of 100,000 IU to patients diagnosed with cancer. This research scarcity could be attributed to the predominant focus on targeted anti-cancer treatments, often overshadowing the consideration for adjunctive therapies or supplementation, such as VD3.

In an RCT reported by Johansson et al., [45] patients with melanoma cancer (II stage) were given to supplement HDVD3. The patients were divided into two groups- the VD3 group, receiving 100,000 IU VD3 every 50 days, and the placebo group. Researchers investigated 25(OH)D levels during three years of supplementation and assessed the effect of VD3 on recurrence in resected, stage II melanoma patients (disease-free survival). The Breslow score calculated based on tumour depth is one of the most important prognostic factors for clinically localized melanoma [37]. The researchers observed an increasing level of 25(OH)D in the treated group (median 32.9 ng/mL) vs. the placebo group (median 19.1 ng/mL). Twelve months after discontinuation of VD3 administration, subjects with low VD3 and Breslow score ≥ 3 mm had shorter disease-free survival compared to the group with Breslow score < 3 mm and/or high levels of 25(OH)D. In addition, researchers observed that participants with a Breslow thickness ≥ 3 mm at diagnosis experienced a lower increase in 25(OH)D levels from baseline to 12 months and were more prone to relapse than participants with a low Breslow score at diagnosis. Overall, they concluded that administration of 100,000 IU VD3 is safe and well tolerated and seems to be an effective treatment in patients with skin cancer (grade II melanoma).

### 3.8. Mortality-Associated Outcomes

The risk of mortality increases with age and the number of diseases older people suffer from [54,61]. Due to the pleiotropic effect of VD3 and the presence of its receptors in many tissues, VD3 has a wide range of effects, including regulating calcium and phosphate metabolism, normalizing blood glucose levels and immune system function [62], which are all particularly important in the elderly population. Due to the low serum level of 25(OH)D in this age group, the effectiveness of HDVD3 supplementation was considered a valuable treatment option in patients in intensive care units, patients at risk of blood transfusion (low hemoglobin level), as well as with the overall increased risk of mortality. Studies selected in this systematic review reported a positive impact on survival, especially in ICU patients. A significant increase in blood hemoglobin was also observed, which could be associated with reduced mortality. Regarding overall infection incidence and potential resulting mortality, no difference was observed after HDVD3 administration.

All selected trials investigating the effect of HDVD3 on mortality in elderly persons were conducted on ICU patients with a high risk of death [7,8,41,43,44]. In the first trial, [41] a significant decrease in hospital stay was noticed in the treatment group vs. the placebo group, 18 ± 11 days compared to 36 ± 19 days, respectively. Patients in the treatment group received HDVD3 for five days, day by day. No side effects connected with VD3 were reported. A shorter hospital stay can be associated with better patient conditions, higher survival scores, and lower mortality.

In the trial by Rake et al. [7], the researchers observed the number of infections and general mortality during the 2-year trial period by comparing the control group (untreated or placebo) with the HDVD3 group [open-label (RCT- OL) or double blind (RCT DB)]. The overall number of infections during the 2-year trial period did not differ between control and VD3 arms. A slightly higher proportion of patients in the control group had at least one infection than those in the VD3 group (28.0% vs. 26.8%). In addition, more participants with low blood VD3 had at least one infection during the trial period than those allocated to VD3. No effect on overall mortality was observed as a result of taking HDVD3.

Another RCT connected with mortality in the elderly was dedicated to HDVD3 and hemoglobin levels in serum, which may influence mortality, especially in ICU elderly patients, because of the necessity of blood transfusion and its complications [44]. In this trial, the two HDVD3s were compared with a placebo- 250,000 IU (5 × 50,000 IU) and 500,000 IU (5 × 100,000 IU). Plasma VD3 concentrations increased significantly after one week in the groups that received 250,000 IU VD3 and 500,000 IU VD3 (to 45 ± 20 ng/mL and 55 ± 14 ng/mL, respectively). There was no change in the placebo group. These effects were sustained for four weeks. Hemoglobin concentrations increased significantly over time in the group that received 500,000 IU VD3. Compared to the placebo group, those who received 5 × 100,000 IU VD3 demonstrated a significant 8% increase per week in hemoglobin concentration. The actual concentrations were 11.3 and 8.2 g/dL. This change was not seen in the 5 × 50,000 IU VD3. Improving hemoglobin concentration in ICU patients may reduce the frequency and necessity of blood transfusions. These findings suggest that HDVD3 may improve iron metabolism in critically ill patients in a short time, which can be associated with their life expectancy.

It has also been reported that VD3 supplementation before or during COVID-19 was associated with better survival after three months in older adults with COVID-19 [8]. The study reported 76.1% (n = 51 individuals) of participants surviving at least three months in the intervention group compared to only 53.6% (n = 15) in the placebo group. In addition, the intervention group was characterized by a longer survival time. However, the longer survival time was not quantified in this study. The preliminary conclusion of this trial is that VD3 supplementation was associated with better 3-month survival in older COVID-19 patients. Sixty-six patients survived three months after leaving the hospital, while 29 patients died in the intervention group. Mortality in this group (23%) was lower when compared to mortality in the placebo group (46.4%).

### 3.9. Changes in VD3 Serum Levels after High-Dose Treatment

A total of 10 studies were included in the analysis [7,8,9,42,43,45,46,48,49,53]. The observed standardized mean differences ranged from 0.65 to 7.65 ng/mL, with all estimates being positive. The estimated average standardized mean difference based on the random-effects model was 2.60 ng/mL (95% CI: 2.07 to 3.13). Therefore, the average outcome differed significantly from zero (z = 9.60, *p* < 0.001).

According to the Q-test, the true outcomes appear heterogeneous (Q = 197.37, *p* < 0.001, tau^2^ = 0.62, *I*^2^ = 95.4%). A 95% prediction interval for the true outcomes ranged from 0.97 to 4.23. Hence, even though there may be some heterogeneity, the true outcomes of the studies are generally in the same direction as the estimated average outcome.

An examination of the studentized residuals revealed that one study (by Johansson et al. [9]) had a value larger than ± 2.8 and may be a potential outlier in the context of this model. According to Cook’s distances, the same study could be considered overly influential.

The regression test indicated funnel plot asymmetry (*p* < 0.01), but no correlation was found between study effects and corresponding sampling variance (*p* = 0.7184), which suggests no publication bias. The results of the meta-analysis are presented in Figure 2.

## 4. Discussion

This study aimed to examine the effect of very high doses of vitamin D3 (HDVD3 over 100,000 IU) on various health outcomes considering the most common diseases in this population (CVD, lung diseases, immunological system functions, cancer) and problems with bone density, falls, and overall mortality in older adults. In all selected clinical trials, results obtained in the VD3 supplementation group were compared with placebo. Each investigation’s findings (positive, negative- different than expected, or unclear) are presented in Table 2 (the last column).

We found that the 25(OH)D level consistently increased after supplementation to the level adopted as a norm for adults. This effect was statistically significant and confirmed by the results of the meta-analysis. Treatment with single or multiple high doses of vitamin D proved to be effective in increasing serum levels of 25(OH)D both in short- 5 days [41,44] and long-term—over three years [9,46,50] regimen. The highest mean difference in serum level of 25(OH)D between the control and treatment group was noted in the Johansson study [42], where participants were dosed with 100,000 IU every 50 days over three years. Conversely, the l–west effect was shown in the study by de Paula et al. [52], where participants received a single dose of VD3 (100,000 IU) during the eight-week observation period. It must be noted that in this study, some participants had low serum 25(OH)D levels at baseline. However, overall, there was no tendency to achieve higher serum 25(OH)D concentrations in patients treated longer or lower serum concentrations in patients treated shorter with HDVD3.

The elderly population is a great challenge for researchers due to its high heterogeneity. Older people may have many interrelated diseases with outcomes affecting each other. They often take many medicines (polypharmacy). Side effects of some drugs are treated with others, which significantly complicates the treatment process. Also, the burden of cancer is considerably higher in older adults when compared to the younger age group [36]. Therefore, studying the effect of vitamin D3 on health could be a promising approach. Although the serum level of 25(OH)D increased over the study period in all studies, the evidence for the therapeutic effect of this treatment is mixed. The researchers did not observe an increase in bone density or a decrease in non-vertebral fractures in the described studies [48]. The improvement of respiratory parameters in older adults (except smokers) was also not achieved [45]. No effect of normalized 25(OH)D serum level was seen in the RA and immune response following influenza vaccination.

For some clinical trials, the protective effect of HDVD3 supplementation in older adults’ health was noted. These studies included mortality-related outcomes in intensive care [7,8], cardiovascular function, [49,53] and lung function in smokers [47]. The positive effect of treatment was also noted in an extended survival time of people diagnosed with skin cancer (melanoma II) and the reduction in hospitalization time and mortality in patients with COVID-19 [9,21,23,52,63].

Our results also showed that the treatment’s length and frequency significantly affected the outcome, especially regarding serum levels of 25(OH)D. The shortest period reported in the selected studies was one dose of VD3 per day, administered for five days. The most prolonged period was 3.3 years with monthly amounts. Based on the results, using HDVD3 frequently during a short period can significantly contribute to increased serum levels. This has been observed, among others, in patients staying in intensive care units [7,8]. In the case of long-term usage, over three years, the level of 25(OH)D in the serum reached a sufficient level, but the studies did not indicate how long it took to reach the concentration considered the norm. Moreover, when using HDVD3 at short intervals, the actual impact of the dose on the effect can be observed more clearly than when using 100,000 IU over a more extended period. Positive treatment results were noted when VD3 was given in a short period, e.g., five days. The results were unsatisfactory in those trials, where 100,000 IU VD3 was administered once per 15 days, month, or 2–3 months. The probable explanation for this observation is the effect of the cumulative dose of VD3. In a short period, the HDVD3 accumulates fast in the organism, and its benefits manifest in improved health conditions. Taking HDVD3 once per month or even less often means the same as taking around 3300 IU VD3 daily, which has a negligible effect on health outcomes, especially when the recommended intake of VD3 for adults is 4000 IU [50]. This dose may have been insufficient for the frequently severe outcomes described in the RCT.

There are several limitations to this review. The biggest is the high heterogeneity of the included clinical trials. Although in all of these studies, the investigators focused on older people, the age of participants varied significantly; e.g., in one trial, the age was 50 [45]; in another, it was over 88 years [8]. In others, it ranged between 50–84 [9,47,50]. The advanced age of some study participants may be associated with their worse initial state, the additional pharmacotherapy, and potential drug interactions that all affect study outcomes. In some trials, the number of patients at the start and end of the study varied, which might also affect their outcomes.

Another limitation lay in the lack of assessment of the baseline 25(OH)D concentration [44] to use it as a criterion to enroll patients. In all the described studies, initial levels of 25(OH)D differed significantly between participants, which could affect the therapeutic process and health outcome. Furthermore, analyzing results based only on the achieved 25(OH)D concentration may also constitute a significant problem. For instance, conflicting results considering the effectiveness of the HDVD3 treatment were observed depending on whether achieved concentration or intention to treat was used as a base for the analysis [64,65]. There was also variation in the dose of VD3 used as a treatment. For instance, in one RCT, the daily dose of VD3 was at first 50,000 IU and then 100,000 IU [44]. Furthermore, the studies differed widely based on the duration of VD3 administration. All these factors hinder comparisons between RCTs.

Heterogeneity also concerns the examined health outcomes, which hampered the selection of the studies and systematizing them into appropriate and clear categories. Individual subgroups contained studies directly or indirectly relating to the generally assigned category. This is because, among many studies investigating the therapeutical effect of 100,000 IU VD3, only a few are concerned with the same outcomes. This also indicates that researchers should focus their work on specific, well-characterized groups of patients to deliver generalizable conclusions about the therapeutic effect of HDVD3 on particular health outcomes.

Future studies should thus employ a more standardized methodology. This should include a fixed number of patients with comparable age, similar comorbidities, and additional pharmacotherapy, allowing higher homogeneity of the study group, standardized examination time, application of similar VD3 doses during the comparable period, and following similar outcomes.

Finally, the studies included in this meta-analysis described the state of research up to June 2022. In a given research period, too few RCTs with HDVD3 describing relatively small populations were found to draw reliable conclusions that could be further translated into the general elderly population.

In conclusion, using a dose of 100,000 IU VD3 administered at short intervals (e.g., daily) for several days may positively impact patients in intensive care units, significantly shorten their hospital stay, and potentially contribute to a faster improvement in the overall health of these patients. In other cases, it may be difficult to draw clear conclusions when HDVD3 is administered at longer intervals (e.g., once a month). The use of a dose of 100,000 IU in older adults may have a beneficial effect on improving the lung function of former or current smokers. Furthermore, HDVD3 may have a positive impact on improving skin cancer-related parameters. For potential improvements in parameters related to bone fractures, bone density, falls, and lower blood pressure in patients with type II diabetes or asthma, the results were inconclusive. Further research and more selective analysis of individual groups are needed to draw reliable conclusions about the impact of HDVD3 on the health of the elderly population.

## Figures and Tables

**Figure 1 nutrients-16-00252-f001:**
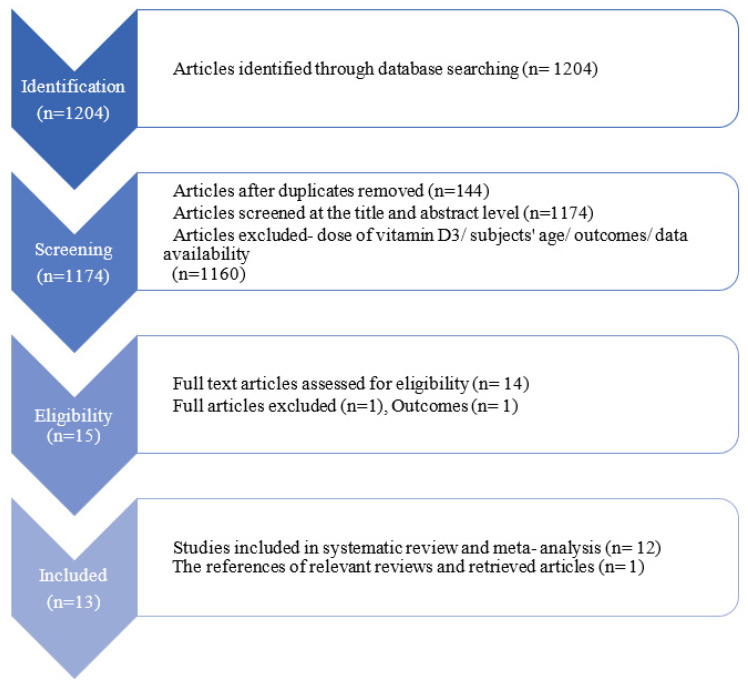
Preferred reporting items for systematic review and meta-analyses (PRISMA) flowchart presenting the study selection process.

**Figure 2 nutrients-16-00252-f002:**
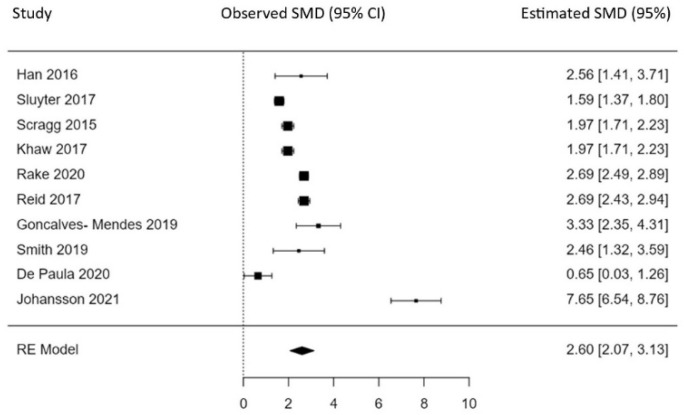
Forest plot for the effect of HDVD supplementation (100,000 IU) on blood serum level in older adults [7,8,9,42,43,45,46,48,49,53]. Squares represent observed standardized mean difference (SMD) with 95% CI. Estimated standardized mean difference with 95% CI calculated based on random effect model.

**Table 1 nutrients-16-00252-t001:** Demographic characterization of the selected studies.

Country	Trials No.	Patients in All Conducted Studies	References
France	3	192	[8,43,52]
UK	1	1901	[46]
USA	2	42	[7,8]
New Zealand	5	5108	[42,45,47,48,49]
Brazil	1	43	[53]
Italy	1	104	[9]

**Table 2 nutrients-16-00252-t002:** Changes in 25(OH)D level at baseline and post supplementation during different time points.

Author, Year	Country	Study Design	Participants (T ^a^/C ^a^)	Age (y)	Baseline VD3 Level (ng/mL) Mean ± SD	Post VD3 Supplementation (ng/mL) Mean ± SD	Study Duration	Dose & Frequency	Results
100,000 IU every day during 5 days									
Han et al., 2016 [7]	USA	DB ^a^, RCT	11/10	63.1	C: 21.5 ± 12.2 T: 20.0 ± 7.3	C: 21 ± 11.2 T: 55 ± 14	5 days	100,000 IU every day	Positive
Smith et al., 2018 [8]	USA	RCT	10/11	C: 64.8 ± 17.5 T: 68.1 ± 18.6	C: 21.5 ± 12.2 T: 20.0 ± 7.3	C: 21.5 ± 12.2 T: 55 ± 14	5 days	100,000 IU for 5 days	Positive
100,000 IU once									
De Paula et al., 2020 [53]	Brazil	RCT	21/22	65 ± 9	C: 14.5 ± 4.3 T: 14.0 ± 5	C: 19.0 ± 5 T: 23.0 ± 7	8 weeks	100,000 IU single dose	Positive
100,000 IU once per 15 days									
Goncalves-Mendes et al., 2019 [43]	France	RCT	19/19	64–77	C: 19.7 ± 5.9 T: 20.7 ± 5.7	C: 18.1 ± 6.7 T: 44.3 ± 8.6	3 months	100,000 IU per 15 days	Negative-different than expected
100,000 IU per month									
Sluyter et al., 2017 [45]	New Zealand	DB, RCT	226/216	50–84	C:24.56 ± 9.48 ^a^* T:26.4 ± 9.76 ^a^*	C: 24.56 ± 9.48 * T: 47.6 ± 18.0 *	1.1 year	100,000 IU per month	Negative-different than expected
Camargo et al., 2021 [47]	New Zealand	RCT	373/402	66.6 ± 8.3	C: 24.16 * T: 25.8 *	C: 24.16 * T: 54.0 *	3.3 years	100,000 IU monthly	Positive in patients who are or have been ever smoker
Scragg et al., 2016 [42]	New Zealand	DB ^a^, RCT	2558/2550 ***^a^ 171/163 ***^a^	50–84	C:24.4 ± 9.6 * T: 24.4 ± 9.6 *	C:26.4 ± 11.6 * T: 54.1 ± 16.0 *	3.3 years	100,000 IU per month	Negative-different than expected
Khaw et al., 2017 [48]	New Zealand	DB, RCT	2558/2550 ***^a^ 171/163 ***^a^	50–84	C:24.4 ± 9.6 * T: 24.4 ± 9.6 *	C:26.4 ± 11.6 * T: 54.1 ± 16.0 *	3.3 years	100,000 IU per month	Negative-different than expected
Rake et al., 2020 [46]	UK	OL ^a^ RCT, DB RCT	372/366 395/392	65–84	C:20.6 ± 5.117 * T: 20.6 ± 5.076 *	C: 20.72 ± 7.647 * T: 43.84 ± 9.435 *	2 years	100,000 IU per month **	Negative-different than expected
Reid et al., 2017 [49]	New Zealand	RCT	228/224	50–84	C:22.4 ± 8.8 * T:22.0 ± 9.2 *	C: 24.0 ± 9.2 * T: 51.6 ± 11.6 *	2 years	100,000 IU per month	Negative-different than expected
100,000 IU per 50 days									
Johansson et al., 2021 [9]	Italy	RCT	52/52 (start study) 25/22 (after 3 years)	50	C: 18.48 ± 1.843 T: 17.97 ± 1.983	C: 22.422 ± 2.15 T: 40.472 ± 2.583	3 years	100,000 IU every 50 days	Positive
100,000 IU per 2 or 3 months									
Annweiler et al., 2021 [52]	France	QE ^a^	67/28	88.0 ± 5.5	C: 29.56 ± 12.84 * T: 24.64 ± 14.16 *	C: NA T: NA	2–3 month	100,000 IU per 2–3 months	Negative-different than expected

^a^ Abbreviations: **T** = tested group, **C** = Control group, **VD3** = vitamin D, **DB** = double blind, **OL** = open label, **RCT** = randomized clinical trial, **NA** = not available, **QE** = quasi-experimental, * VD3 in the original article was presented in nmol/L; it was recalculated, calculation: 1 ng/mL = 2.496 nmol/L, but in calculations, 2.5 nmol/l was taken; ** 100,000 IU vitamin D was administered for patients allocated into OL RCT and DB RCT; *** the first row is number of participant at the beginning of the study, the second number (low row) is the number of participants at the end of the study that was taken to the calculation.

**Table 3 nutrients-16-00252-t003:** Risk of bias assessment includes studies using the Cochrane Collaboration tool across seven domains. Risk of bias levels: low (“+”), unclear (“?”), high (“-”).

	Bias	Random Sequence Generation (Selection Bias)	Allocation Concealment (Selection Bias)	Blinding Participants and Personnel (Performance Bias)	Incomplete Outcome Data (Attrition Bias)	Selective Reporting (Reporting Bias)	Other Bias
References							
Han et al. [7]		+	+	+	+	+	?
Soubrier et al. [44]		?	?	-	+	+	?
Sluyter et al. [45]		+	+	+	+	+	?
Annweiler et al. [52]		-	?	-	_+_	+	?
Scragg et al. [42]		+	+	+	+	+	?
Khaw et al. [48]		+	+	+	+	+	?
Rake et al. [46]		+	+	+	+	+	?
Reid et al. [49]		?	?/+	?	+	+	?
Goncalves- Mendes et al. [43]		+	+	?	+	+	?
Smith et al. [8]		+	+	+	+	+	?
De Paula et al. [53]		+	+	+	+	+	?
Camargo et al. [47]		+	?	?	+	+	?
Johansson et al. [9]		+	+	+	+	+	?

**Table 4 nutrients-16-00252-t004:** Characteristics of the included studies.

Author, Year	Country	Study Design	Participants (T ^a^/C ^a^)	Age (y)	Baseline 25(OH)D Level (ng/mL) Mean ± SD	Post VD3 Supplementation Level of 25(OH)D (ng/mL) Mean ± SD	Study Duration	Supplementation	Dose & Frequency	Disease/Outcome
Han et al., 2016 [7]	USA	DB ^a^, RCT	11/10	63.1	C: 21.5 ± 12.2 T: 20.0 ± 7.3	C: 21 ± 11.2 T: 55 ± 14	5 days	VD3	100,000 IU every day	VICU ^a^ patients
Soubrier et al., 2018 [8]	France	DB, RCT	29/30	59.8 ± 10.9	C: NA ^a^ T: NA	C: NA T: NA	24 weeks	VD3	100,000 IU per 4 weeks	RA ^a^
Sluyter et al., 2017 [45]	New Zealand	DB, RCT	226/216	50–84	C: 24.56 ± 9.48 ^a^* T: 26.4 ± 9.76 ^a^*	C: 24.56 ± 9.48 * T: 47.6 ± 18.0 *	1.1 year	VD3	100,000 IU per month	lung function
Annweiler et al., 2021 [52]	France	QE ^a^	67/28	88.0 ± 5.5	C: 29.56 ± 12.84 * T: 24.64 ± 14.16 *	C: NA T: NA	2-3 month	VD3	100,000 IU per 2-3 months	COVID-19
Scragg et al., 2016 [42]	New Zealand	DB ^a^, RCT	2558/2550 ***^a^ 171/163 ***^a^	50–84	C: 24.4 ± 9.6 * T: 24.4 ± 9.6 *	C:26.4 ± 11.6 * T: 54.1 ± 16.0 *	3.3 years	VD3	100,000 IU per month	CD ^a^
Khaw et al., 2017 [48]	New Zealand	DB, RCT	2558/2550 ***^a^ 171/163 ***^a^	50–84	C: 24.4 ± 9.6 * T: 24.4 ± 9.6 *	C:26.4 ± 11.6 * T: 54.1 ± 16.0 *	3.3 years	VD3	100,000 IU per month	falls, NVF ^a^
Rake et al., 2020 [46]	UK	OL ^a^ RCT, DB RCT	372/366 395/392	65–84	C: 20.6 ± 5.117 * T: 20.6 ± 5.076 *	C: 20.72 ± 7.647 * T: 43.84 ± 9.435 *	2 years	VD3	100,000 IU per month **	mortality in people aged 65-84 years
Reid et al., 2017 [49]	New Zealand	RCT	228/224	50–84	C: 22.4 ± 8.8 * T: 22.0 ± 9.2 *	C:24.0 ± 9.2 * T:51.6 ± 11.6 *	2 years	VD3	100,000 IU per month	BD ^a^
Goncalves- Mendes et al., 2019 [43]	France	RCT	19/19	64–77	C: 19.7 ± 5.9 T: 20.7 ± 5.7	C: 18.1 ± 6.7 T: 44.3 ± 8.6	3 months	VD3	100,000 IU per 15 days	influenza vaccine response
Smith et al., 2018 [8]	USA	RCT	10/11	C:64.8 ± 17.5 T:68.1 ± 18.6	C: 21.5 ± 12.2 T: 20.0 ± 7.3	C: 21.5 ± 12.2 T: 55 ± 14	5 days	VD3	100,000 IU for 5 days	hemoglobin concentration in MVICP ^a^
De Paula et al., 2020 [53]	Brazil	RCT	21/22	65 ± 9	C: 14.5 ± 4.3 T: 14.0 ± 5	C: 19.0 ± 5 T: 23.0 ± 7	8 weeks	VD3	100,000 IU single dose	BP ^a^ in patients with hypertension, 2 DM ^a^ and hypovitaminosis D3
Camargo et al., 2021 [47]	New Zealand	RCT	373/402	66.6 ± 8.3	C: 24.16 * T: 25.8 *	C: 24.16 * T: 54.0 *	3.3 years	VD3	100,000 IU monthly	asthma and/or COPD ^a^
Johansson et al., 2021 [9]	Italy	RCT	52/52 (start study) 25/22 (after 3 years)	50	C: 18.48 ± 1.843 T: 17.97 ± 1.983	C: 22.422 ± 2.15 T: 40.472 ± 2.583	3 years	VD3	100,000 IU every 50 days	disease-free survival in stage II melanoma
Han, 2016 [7]	USA	DB ^a^, RCT	11/10	63.1	C: 21.5 ± 12.2 T: 20.0 ± 7.3	C: 21 ± 11.2 T: 55 ± 14	5 days	VD3	100,000 IU every day	VICU ^a^ patients
Soubrier, 2018 [8]	France	DB, RCT	29/30	59.8 ± 10.9	C: NA ^a^ T: NA	C: NA T: NA	24 weeks	VD3	100,000 IU per 4 weeks	RA ^a^
Sluyter, 2017 [45]	New Zealand	DB, RCT	226/216	50–84	C:24.56 ± 9.48 ^a^* T:26.4 ± 9.76 ^a^*	C: 24.56 ± 9.48 * T: 47.6 ± 18.0 *	1.1 year	VD3	100,000 IU per month	lung function
Annweiler, 2021 [52]	France	QE ^a^	67/28	88.0 ± 5.5	C:29.56 ± 12.84 * T:24.64 ± 14.16 *	C: NA T: NA	2-3 month	VD3	100,000 IU per 2-3 months	COVID-19
Scragg, 2016 [42]	New Zealand	DB ^a^, RCT	2558/2550 ***^a^ 171/163 ***^a^	50–84	C:24.4 ± 9.6 * T: 24.4 ± 9.6 *	C:26.4 ± 11.6 * T: 54.1 ± 16.0 *	3.3 years	VD3	100,000 IU per month	CD ^a^
Khaw, 2017 [48]	New Zealand	DB, RCT	2558/2550 ***^a^ 171/163 ***^a^	50–84	C:24.4 ± 9.6 * T: 24.4 ± 9.6 *	C:26.4 ± 11.6 * T: 54.1 ± 16.0 *	3.3 years	VD3	100,000 IU per month	falls, NVF ^a^
Rake, 2020 [46]	UK	OL ^a^ RCT, DB RCT	372/366 395/392	65–84	C:20.6 ± 5.117 * T: 20.6 ± 5.076 *	C: 20.72 ± 7.647 * T: 43.84 ± 9.435 *	2 years	VD3	100,000 IU per month **	mortality in people aged 65-84 years
Reid, 2017 [49]	New Zealand	RCT	228/224	50–84	C:22.4 ± 8.8 * T:22.0 ± 9.2 *	C:24.0 ± 9.2 * T:51.6 ± 11.6 *	2 years	VD3	100,000 IU per month	BD ^a^
Goncalves- Mendes, 2019 [43]	France	RCT	19/19	64–77	C: 19.7 ± 5.9 T: 20.7 ± 5.7	C: 18.1 ± 6.7 T: 44.3 ± 8.6	3 months	VD3	100,000 IU per 15 days	influenza vaccine response
Smith, 2018 [8]	USA	RCT	10/11	C:64.8 ± 17.5 T:68.1 ± 18.6	C: 21.5 ± 12.2 T: 20.0 ± 7.3	C: 21.5 ± 12.2 T: 55 ± 14	5 days	VD3	100,000 IU for 5 days	hemoglobin concentration in MVCIP ^a^
De Paula, 2020 [53]	Brazil	RCT	21/22	65 ± 9	C: 14.5 ± 4.3 T: 14.0 ± 5	C: 19.0 ± 5 T: 23.0 ± 7	8 weeks	VD3	100,000 IU single dose	BP ^a^ in patients with hypertension, 2 DM ^a^ and hypovitaminosis D3
Camargo, 2021 [47]	New Zealand	RCT	373/402	66.6 ± 8.3	C: 24.16 * T: 25.8 *	C: 24.16 * T: 54.0 *	3.3 years	VD3	100,000 IU monthly	asthma and/or COPD ^a^
Johansson, 2021 [9]	Italy	RCT	52/52 (start study) 25/22 (after 3 years)	50	C: 18.48 ± 1.843 T: 17.97 ± 1.983	C: 22.422 ± 2.15 T: 40.472 ± 2.583	3 years	VD3	100,000 IU every 50 days	disease-free survival in stage II melanoma

^a^ Abbreviations: **T** = tested group, **C** = Control group, **VD3** = vitamin D, **DB** = double blind, **OL** = open label, **RCT** = randomized clinical trial, **NA** = not available, **QE** = quasi-experimental, **BD** = bone density, **BP** = blood pressure, **COPD** = Chronic Obstructive Pulmonary Disease, **VICU** = Ventilated Intensive Care Units Patients, **RA** = rheumatoid arthritis, **CD** = cardiovascular disease, **NVF** = non- vertebral fractures, **MVCIP** = Mechanically Ventilated Critically Ill Patients; * VD3 in the original article was presented in nmol/L; it was recalculated, calculation: 1 ng/ml= 2.496 nmol/l, but in calculations, 2.5 nmol/L was taken; ** 100,000 IU vitamin D was administered for patients allocated into OL RCT and DB RCT; *** the first row is number of participants at the beginning of the study, the second number (low row) is the number at the end of the study that was taken to the calculation.

## Data Availability

Data described in the manuscript will be made available upon request pending application and approval.

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
