# Peer review of "Has a High Dose of Vitamin D3 Impacted Health Conditions in Older Adults?—A Systematic Review and Meta-Analysis Focusing on Dose 100,000 IU"

_nutrients, 2024, doi:10.3390/nu16020252_

Round 1

Reviewer 1 Report

Comments and Suggestions for Authors

The abstract has 536 words. That is too many, 250-300 would be much better. The abstract need not go into all the details; just enough that readers get the idea and then decide to read the paper.

The limitations of RCTs should be discussed. One of the main ones is not measuring baseline 25(OH)D concentration and using that as the basis for including participants as well as analyzing outcomes based on achieved 25(OH)D concentration.

Comparing the Evidence from Observational Studies and Randomized Controlled Trials for Nonskeletal Health Effects of Vitamin D.

Grant WB, Boucher BJ, Al Anouti F, Pilz S.Nutrients. 2022 Sep 15;14(18):3811. doi: 10.3390/nu14183811.PMID: 36145186 Free PMC article. Review.

Critical Appraisal of Large Vitamin D Randomized Controlled Trials.

Pilz S, Trummer C, Theiler-Schwetz V, Grübler MR, Verheyen ND, Odler B, Karras SN, Zittermann A, März W.Nutrients. 2022 Jan 12;14(2):303. doi: 10.3390/nu14020303.

The original analysis of the D2d trial found no effect based on intention to treat.

Vitamin D Supplementation and Prevention of Type 2 Diabetes.

Pittas AG, Dawson-Hughes B, Sheehan P, Ware JH, Knowler WC, Aroda VR, Brodsky I, Ceglia L, Chadha C, Chatterjee R, Desouza C, Dolor R, Foreyt J, Fuss P, Ghazi A, Hsia DS, Johnson KC, Kashyap SR, Kim S, LeBlanc ES, Lewis MR, Liao E, Neff LM, Nelson J, O'Neil P, Park J, Peters A, Phillips LS, Pratley R, Raskin P, Rasouli N, Robbins D, Rosen C, Vickery EM, Staten M; D2d Research Group.N Engl J Med. 2019 Aug 8;381(6):520-530. doi: 10.1056/NEJMoa1900906. 

However, an analysis based on achieved 25(OH)D concentration in the vitamin D treatment arm did

Intratrial Exposure to Vitamin D and New-Onset Diabetes Among Adults With Prediabetes: A Secondary Analysis From the Vitamin D and Type 2 Diabetes (D2d) Study.

Dawson-Hughes B, Staten MA, Knowler WC, Nelson J, Vickery EM, LeBlanc ES, Neff LM, Park J, Pittas AG; D2d Research Group.Diabetes Care. 2020 Dec;43(12):2916-2922. doi: 10.2337/dc20-1765. 

COVID-19 – from a search at Google Scholar

A single-oral bolus of 100,000 IU of cholecalciferol at hospital admission did not improve outcomes in the COVID-19 disease: The COVID-VIT-D—A …

…, JL Fernández-Martín, COVID-VIT-D trial collaborators - BMC medicine, 2022 - Springer

Positive effects of vitamin D supplementation in patients hospitalized for COVID-19: a randomized, double-blind, placebo-controlled trial

…, M Trémège, M Coffiner, AF RousseauCalmes… - Nutrients, 2022 - mdpi.com

… of COVID-19 patients requiring hospitalization was improved by administration of vitamin
D. … Subjects were administered either placebo, 50,000 IU vitamin D or 100,000 IU vitamin D …

Effect of cholecalciferol supplementation on the clinical features and inflammatory markers in hospitalized COVID-19 patients: a randomized, open-label …

TL Karonova, KA Golovatyuk, IV Kudryavtsev… - Nutrients, 2022 - mdpi.com

… vitamin D status of 311 patients hospitalized with COVID-19 (… 100,000 IU cholecalciferol
supplementation in addition to standard 
COVID-19 therapy led to an increase in serum 25(OH)D 

COVID19 and Vitamin D: A lesson from the skin

RM Slominski, J Stefan, M Athar… - Experimental …, 2020 - Wiley Online Library

… Vitamin D toxicity is typically not observed until extremely high doses of vitamin D in the range
of 50,000-
100,000 IUs … [ 70 ] Doses up to 500 000 IUs have been routinely given to nursing …

Possibly of interest for the intro or discussion

Daily oral dosing of vitamin D3 using 5000 TO 50,000 international units a day in long-term hospitalized patients: Insights from a seven year experience.

McCullough PJ, Lehrer DS, Amend J.J Steroid Biochem Mol Biol. 2019 May;189:228-239. doi: 10.1016/j.jsbmb.2018.12.010. 

A search of pubmed.gov with “Amrein, vitamin d, high-dose” found 16 entries. While the manuscript focused on 100,000 IU administered once, there should at least be a discussion of the higher doses used by Amrien et al. Perhaps the title should be modified to include 100,000 IU once.

The estimated average stand-ardised mean difference based on the random-effects model was 2.60 (95% CI: 2.07 to 3.13) suggest- ing consistent positive effect of the treatment,.

Comment: It is not readily apparent what this means.

Scientific research shows that a proper level of VD3

Comment: You cannot use the same symbol for vitamin D3 and 25-hydroxyvitamin D3. For the second, use 25-hydroxyvitamin D3 [25(OH)D] at first time in the abstract and text, 25(OH)D thereafter.

Scientific research shows that a proper level of VD3 in blood serum maintains appro-

90

priate mineral balance and reduces the risk of diseases such as type 2 diabetes [8-9] or

91

depression.[10-11] Moreover, VD3 treats many diseases, such as osteomalacia in adults,

92

[12] thyroid and autoimmunological diseases, [13] rheumatoid arthritis, and Hashimoto

93

disease.

Consider adding:

Intratrial Exposure to Vitamin D and New-Onset Diabetes Among Adults With Prediabetes: A Secondary Analysis From the Vitamin D and Type 2 Diabetes (D2d) Study.

Dawson-Hughes B, Staten MA, Knowler WC, Nelson J, Vickery EM, LeBlanc ES, Neff LM, Park J, Pittas AG; D2d Research Group.Diabetes Care. 2020 Dec;43(12):2916-2922. doi: 10.2337/dc20-1765. 

Effect of Vitamin D Supplements on Relapse or Death in a p53-Immunoreactive Subgroup With Digestive Tract Cancer: Post Hoc Analysis of the AMATERASU Randomized Clinical Trial.

Kanno K, Akutsu T, Ohdaira H, Suzuki Y, Urashima M.JAMA Netw Open. 2023 Aug 1;6(8):e2328886. doi: 10.1001/jamanetworkopen.2023.28886.

Ref. 11 does not deal with depression.

See, instead

Vitamin D and depression: a critical appraisal of the evidence and future directions

V Menon, SK Kar, N Suthar… - Indian journal of …, 2020 - journals.sagepub.com

Efficacy of vitamin D supplementation in major depression: A meta-analysis of randomized controlled trials

F Vellekkatt, V Menon - Journal of postgraduate medicine, 2019 - ncbi.nlm.nih.gov

  References: Generally the names of all authors of each reference are included.

Comments on the Quality of English Language

It seems ok.

Author Response

Dear reviewer,

Please look at the answers to your comments in the attached file.

Reviewer 2 Report

Comments and Suggestions for Authors

The authors of this paper have carried out a meta-analysis of studies where older persons have been given supplements of vitamin D. They evaluated the effects of the supplement on blood levels of vitamin D and on health. There has been much interest in recent years on the possible relationship between vitamin D status and health. Many studies have investigated the possible benefits of supplements of the vitamin. For that reason the findings of this paper are potentially of much interest.

The meta-analysis appears to have been well carried out and the findings provide much information that appears to be of value.

Unfortunately, the paper is written in poor English. The authors should have had the paper edited by someone with strong skills at scientific English. I would have been fully justified in rejecting the paper and insisting on the paper being properly edited and then resubmitted. But instead of doing that I spent about 6 hours editing the paper. In order to do that I converted the paper to Word and then made the corrections using tracking. I have made around 200 corrections. I am quite sure that there are dozens more problems with the writing that I failed to notice.

It is essential that the authors copy my corrections into a new draft. The paper should then be sent for additional editing by someone with strong skills at scientific English.

Comments on the Quality of English Language

see comments above.

The Word document with my comments in tracking is attached. Please send it to the authors.

Author Response

(The authors gave the same response as above.)
